# Advances in the Pathogenesis of Steroid-Associated Osteonecrosis of the Femoral Head

**DOI:** 10.3390/biom14060667

**Published:** 2024-06-06

**Authors:** Jie Zhang, Jianze Cao, Yongfei Liu, Haiyan Zhao

**Affiliations:** 1The First Clinical College of Medicine, Lanzhou University, Lanzhou 730000, China; 220220907181@lzu.edu.cn (J.Z.); 220220907091@lzu.edu.cn (J.C.); 220220907161@lzu.edu.cn (Y.L.); 2Department of Orthopedics, The First Hospital of Lanzhou University, Lanzhou 730000, China

**Keywords:** steroid-associated osteonecrosis of the femoral head, pathogenesis, glucocorticoid

## Abstract

Osteonecrosis of the femoral head (ONFH) is a refractory orthopedic condition characterized by bone cell ischemia, necrosis, bone trabecular fracture, and clinical symptoms such as pain, femoral head collapse, and joint dysfunction that can lead to disability. The disability rate of ONFH is very high, which imposes a significant economic burden on both families and society. Steroid-associated osteonecrosis of the femoral head (SANFH) is the most common type of ONFH. However, the pathogenesis of SANFH remains unclear, and it is an urgent challenge for orthopedic surgeons to explore it. In this paper, the pathogenesis of SANFH and its related signaling pathways were briefly reviewed to enhance comprehension of the pathogenesis and prevention of SANFH.

## 1. Introduction

ONFH is a common and refractory disease in orthopedics, characterized by challenging early diagnosis and a high disability rate. China has the highest number of patients with ONFH. The pathogenic factors of femoral head necrosis include glucocorticoids (GCs), alcohol, hyperlipidemia, and other factors. Excessive use of glucocorticoids is the primary cause of ONFH [1]. The incidence of SANFH increases annually due to the high doses and long-term use of glucocorticoids in the clinical treatment of rheumatic diseases, immune-related diseases, spinal shock, and other conditions. It has become the primary cause of femoral head necrosis. Due to the excessive use of glucocorticoids, the incidence of femoral head necrosis has been increasing worldwide in recent years. According to an epidemiological survey, the number of patients with femoral head necrosis in Japan has reached 25,000 [2], and the number of patients aged 15 and above in China has exceeded 8 million. The disability rate of this disease is very high. If there is no early intervention for SANFH, the collapse rate of the femoral head structure can exceed 80% within 2 years, imposing a significant burden on patients and their families [3]. Due to the irreplaceability of GCs in clinical treatment, the pathogenesis of SANFH must be continuously explored to provide a theoretical basis for the prevention and treatment of SANFH. With the deepening of research, the pathogenesis of SANFH is gradually becoming clearer. In this paper, the pathogenesis of SANFH is summarized to provide a basis for the prevention and treatment of SANFH.

## 2. The Dose and Time Effect Relationship of SANFH

ONFH is often caused by glucocorticoids, which is one of the most common non-traumatic causes. However, due to the various pathogenic factors of ONFH, determining the specific timing and dosage of glucocorticoids that can lead to hormone-induced femoral head necrosis remains a challenging issue. Studies have shown that a cumulative dose of glucocorticoids ranging from 1800 to 15,505 mg prednisone or its glucocorticoid equivalent can lead to the occurrence of ONFH [4]. One study found a significant association between the use of GCs at a dosage of 40 mg/day and osteonecrosis, with an increase of around 4% for every 10 mg/day [5]. Glucocorticoids (GCs) were considered to be the causative factor of ONFH in patients whose cumulative GCs dose reached 2 g within 3 months and whose last use of GCs was within 2 months [6]. In addition, the method of administering GCs is also associated with the occurrence of ONFH. Systemic administration is more likely to cause hormonal femoral head necrosis than local administration. Additionally, intra-nodule injection in patients is associated with hormonal femoral head necrosis. However, the pathogenesis of SANFH is very complex and may be closely related to the following mechanisms.

## 3. Relative Mechanisms of SANFH

### 3.1. Endothelial Cell Damage, Coagulation Abnormalities, and Tendency for Thrombosis

Because of its unique anatomical structure, the femoral head has long blood vessels and a large blood supply range. The distribution of blood vessels is small and weak, making it difficult to form collateral circulation. This can lead to inadequate blood supply and the formation of micro-thrombi. As an essential component of the bone microenvironment, the bone microvascular system serves not only to support the metabolic requirements of various bone cells but also to facilitate secretion. Microcirculation dysfunction can affect new bone formation, bone absorption, nutrient transport, and the balance of the bone microenvironment [7].

During the process of SANFH, abnormalities in thrombosis/thrombin and dysfunction of endothelial cells result in disturbances in bone tissue microcirculation. This leads to a chain reaction of bone hypoxia and nutrient deficiency, weakening or even eliminating the pressure resistance of the femoral head, ultimately causing varying degrees of femoral head collapse. Endothelial cells, mainly bone microvascular endothelial cells (BMECs) and endothelial progenitor cells (EPCs), can differentiate into BMECs. BMECs are distributed on the inner surface of bone microvessels and bone sinuses, maintaining the local blood supply of the femoral head. They regulate the contraction and relaxation of blood vessels by secreting vasoactive substances. This regulation forms the basis for the exchange of bone microcirculation substances, stabilizes blood flow, and participates in the formation of new bone and blood vessels [8].

The decrease in blood supply to the femoral head is a crucial factor in the development of femoral head necrosis. Therefore, damage to bone marrow endothelial cells (BMECs) may be the key element in the onset of femoral head necrosis. Sustained high doses of GCs can induce endothelial cell dysfunction. Vascular endothelial cells form the inner layer of blood vessel walls and play a crucial role in maintaining vascular homeostasis. The integrity of endothelial cells is essential for maintaining blood flow in the body. Endothelial cells are constantly exposed to inflammatory and stressful environments, which can lead to endothelial cell damage and the secretion of various substances that regulate the clotting process.

Large doses of GCs can induce platelet aggregation and endothelial cell apoptosis, which stimulates the binding of platelets to endothelial cells, leading to platelet activation and thrombosis [9]. Endothelial cells can produce heparan sulfate and other heparan-like proteoglycans, exerting an anticoagulant effect by adhering to antithrombin III and inactivating thrombin and Xa. The patients with ONFH showed significantly increased platelet activation. Histopathological observations revealed platelet thrombosis in micro-vessels near the necrotic areas [10], which may be secondary to BMEC injury caused by GCs.

High doses of dexamethasone inhibit fibrinolytic activity by decreasing t-PA activity and increasing PAI-1 antigen levels. PAI-1 forms a complex with t-PA to participate in fibrinolysis. GCs increase the activity of PAI-1, leading to reduced fibrinolysis and a relatively high coagulation state [11]. The A2M protein is located on the luminal surface of endothelial cells and functions to inhibit plasmin and kallikrein. It is an inhibitor of fibrinolysis. In a rat model of ONFH induced by GCs, the gene expression of the A2M protein on the surface of the endothelial lumen was significantly upregulated [12]. Therefore, GCs can alter BMEC function by regulating alpha-2-macroglobulin expression. Studies have shown that Methylprednisolone (MPS) can cause damage and dysfunction to BMECs by regulating the formation of cytokines, cytochrome C, pro-apoptotic protein (Bad)/anti-apoptotic protein (Bcl-xL) complexes, and increasing reactive oxygen species (ROS). This significantly inhibits angiogenesis, leading to abnormal blood supply to the femoral head [13]. BMECs regulate the local tension of blood vessels by producing and releasing vasoactive substances such as nitric oxide (NO) and prostacyclin-I2 (PGI-2). 6-ketoprostacycline F1 is considered a marker of endothelial cell injury and a metabolite of PGI-2 [14]. The level of 6-ketoprostacyclin F1 was significantly reduced in an SANFH rabbit model, suggesting that GCs may cause injury to BMECs through this mechanism, thereby contributing to the development of SANFH.

GCs can induce the overexpression of ligation-mediated regulatory proteins (JMY) in endothelial cells, leading to the upregulation of Bax and VE-cad in endothelial cells. This process induces endothelial cell injury and decreases endothelial cell motility [15]. In SANFH model rats, plasma viscosity, blood lipids, and pro-inflammatory cytokines were increased. The blood was in a hypercoagulable state, with a significantly increased tendency for thrombosis. Additionally, the number of new vessels was decreased, and bone integrity was significantly damaged [16].

### 3.2. Oxidative Stress and Reactive Oxygen Species Mechanism

Oxidative stress refers to the imbalance between the production of reactive oxygen species in cells and the antioxidant capacity of cells, leading to the disruption of the redox balance and initiating a cascade of biological reactions. This reaction may lead to oxidative damage of biomolecules such as proteins, lipids, and nucleic acids, which can result in cell damage and even cell death. Glucocorticoids can promote the production of free radicals in several ways, including by increasing the activity and membrane permeability of the mitochondrial respiratory chain. This action causes the mitochondria to produce more oxygen free radicals [17].

A large dose of GCs can induce cellular oxidative stress, leading to mitochondrial damage and the release of a large number of ROS, as well as an imbalance in the endogenous oxidative defense mechanism [18]. Excessive ROS can disrupt the oxidant–antioxidant balance in mitochondria, leading to oxidative stress. This imbalance can cause cellular and blood vessel dysfunction, ultimately resulting in compromised blood flow and necrosis of the femoral head. High doses of GCs can impair the mitochondrial function of BMSCs and osteoblasts (OBs), leading to elevated levels of ROS and reduced antioxidant levels, such as superoxide dismutase (SOD) and glutathione peroxidase (GSH-Px). This results in the occurrence of oxidative stress, which then inhibits the classical osteogenic signaling pathways such as Wnt/β-catenin and MAPK, leading to impaired function and cell viability of BMSCs and OB, ultimately resulting in the occurrence of SANFH [19].

In addition, GCs can also activate the oxidase system, further increasing the generation of free radicals. Glucocorticoids may inhibit the function of the antioxidant defense system by inhibiting the activity of antioxidant enzymes and reducing the synthesis of antioxidant vitamins. This leads to a decrease in antioxidant capacity within cells and an inability to effectively combat oxidative stress. Glucocorticoids can enhance cell sensitivity to oxidative stress, rendering cells more susceptible to oxidative damage. Glucocorticoids may impact the function of the DNA repair system, hindering the repair of DNA damage and consequently increasing the sensitivity of cells to oxidative stress. Osteoblasts, osteocytes, and osteoclasts cooperate to maintain the balance of bone metabolism in normal bone tissues. BMSCs can differentiate into osteoblasts to maintain the balance between osteoblasts and osteoclasts [20]. Oxidative stress directly affects the functions of BMSCs, osteoblasts, and osteoclasts, leading to the disruption of bone metabolic balance and the onset and development of ONFH [21].

Oxidative stress can reduce the activity of BMSCs, accelerate their senescence, and induce their apoptosis. However, the activity of BMSCs is significantly improved after the use of antioxidant drugs to remove reactive oxygen species [22]. Moreover, reactive oxygen species can slow down the cell cycle and weaken the proliferation of BMSCs [23]. A large number of studies have shown that GCs can not only induce monoacylglycerol lipase (MAGL) to inhibit the activation of the antioxidant pathway keap1/Nrf2 but also cause high oxidative stress in BMSCs, leading to cell death. GCs lead to the down-regulation of genes encoding antioxidant enzymes such as CAT, NQO1, and HMOX1 in necrotic bone. This leads to high oxidative stress in the necrotic area of the femoral head, resulting in a significant increase in related proteins such as Receptor Activator of Nuclear Factor-**κ**B Ligand (RANKL) and cathepsin K. These proteins promote osteoclast differentiation and activation, leading to bone resorption. This process promotes the recruitment and activation of osteoclasts, resulting in increased bone resorption, massive bone loss, and ultimately the occurrence of SANFH [19,24]. Therefore, large doses of glucocorticoids can induce the production of a significant number of reactive oxygen species in BMSCs, leading to oxidative stress. This oxidative stress can impair their activity, proliferation, and osteogenic differentiation. Osteoblasts, which are derived from BMSCs, can regulate bone matrix mineralization, maintain bone mass, promote bone development, and are an important component in maintaining normal bone metabolism [25].

Bone morphogenetic protein-2 (BMP-2) has a strong osteogenic capacity and is a key regulator of bone formation. Decreased levels indicate impaired bone formation. The level of ROS in MC stimulated by Dex increased significantly for an extended period, leading to the dysfunction of osteoblasts. This dysfunction resulted in decreased expression of BMP-2, osteonectin, bone-specific transcription factor (RUNX2), and other specific osteogenic markers with potent osteogenic effects [26]. The oxidative stress induced by GCs impaired the lipid and protein synthesis of osteoblasts, resulting in a prolonged cell cycle, reduced proliferation ability, and impaired osteoblast function [25,27,28]. The synthesis and mineralization of bone matrix require the adhesion of osteoblasts. Excessive ROS can disrupt the adhesion of osteoblasts, leading to bone damage [29]. Liang et al. [30] showed that PGK1 depletion protects human osteoblasts from GCs by activating the Keap1-Nrf2 signaling cascade. Targeting the PGK1-Nrf2 cascade effect may be a novel strategy to protect osteoblasts against GC-induced oxidative damage. Excessive ROS can lead to oxidative stress, which may hinder the function, proliferation, and differentiation of osteoblasts. This can result in osteogenic disorders and compromised bone formation. Bone cells are the most abundant cells in bone tissue and play a crucial role in the process of bone injury and repair. The essence of osteonecrosis is the death of bone cells, and high doses of GCs can trigger apoptosis of bone cells, resulting in the development of SANFH [31].

In the process of bone remodeling, bone cells play a crucial role in bone repair, and their activity is closely linked to bone remodeling. However, oxidative stress induced by high doses of GCs will inhibit the activity of bone cells, which poses an obstacle to bone remodeling and bone injury repair [32]. Kar et al. found that an excess of ROS would lead to a decrease in the autophagy activity of bone cells [33]. Good autophagy activity in cells protects them from oxidative stress, while inhibiting autophagy increases ROS-induced bone cell death. Treatment of bone cells with teriparatide can reduce the high ROS levels caused by GCs and promote the proliferation of bone cells. This confirms the damage to bone cells caused by oxidative stress. Combating oxidative stress can reduce the damage of GCs to bone cells [34]. As an indispensable cell for maintaining normal bone metabolism, osteoclasts are activated by the interaction of RANKL with the receptor activator of nuclear factor κB (RANK), which serves a crucial function. Excessive bone resorption is a significant factor in the development of SANFH [5,35].

Excessive ROS can not only promote the proliferation of osteoclasts but also enhance the maturation and differentiation of osteoclasts, leading to increased bone resorption and decreased bone mass [34,36]. Thymoquinone has an antioxidant effect, which can reduce the expression level of osteoclast-related genes such as carbonic anhydrase II, matrix metalloproteinase 9, and tartrate-resistant acid phosphatase by decreasing the production of ROS. This action leads to impaired osteoclast formation and inhibits osteoclast resorption [37]. The use of antioxidants can reduce osteoclast differentiation and decrease osteoclast activity, thereby reducing bone resorption [32]. The aforementioned studies have confirmed that excessive ROS can lead to increased proliferation and differentiation of osteoclasts, resulting in elevated bone resorption and reduced bone mass. This may be a significant factor in the pathogenesis of SANFH. Bone tissue is composed of BMSCs, osteoblasts, and osteoclasts. Under normal circumstances, all types of cells carry out their specific functions to maintain regular bone metabolism. According to the above review, a high dose of GCs can stimulate various cells to produce excessive ROS, leading to oxidative stress. This disrupts normal bone metabolism and contributes to the development of SANFH, as illustrated in Figure 1.

### 3.3. Lipid Metabolism Disorder and Fat Embolism Mechanism

GCs can cause swelling and necrosis of adipocytes, increased lipid deposition in bone cells, hyperlipidemia, fat embolism, adipogenesis of bone marrow stromal cells, and infiltration of bone marrow fat. This is due to interosseous venous stasis with fat infiltration, leading to the interruption of interosseous microcirculation. Consequently, structural changes in the femoral head occur, resulting in decreased blood flow, further progression of SANFH, and collapse. Moreover, it has been found that a high-fat diet can exacerbate femoral head necrosis. The occurrence of femoral head necrosis is associated with an increase in interleukin-6 (IL-6) secretion in macrophages [38]. Bai pointed out that high triglyceride (TG) and low high-density lipoprotein (HDL-C) levels were both associated with SANFH [39].

In addition, SANFH was associated with higher apolipoprotein-B (Apo-B) levels and the ApoB/A1 ratio. Wang [40] showed that mice treated with GCs for one month had significantly higher levels of serum cholesterol (CHOL) and greater lipid accumulation in the cancellous bone, including the femoral head. It has been suggested that the Runx2 gene is a key regulator of osteogenic differentiation, and GCs can decrease the expression of the Runx2 gene and increase the expression of the adipogenic differentiation factor, namely the peroxisome proliferator-activated receptor-γ (*PPARγ-2*) gene and protein, resulting in increased adipocyte differentiation, abnormal lipid metabolism, and inhibition of osteogenic differentiation [41].

### 3.4. Mechanisms of Apoptosis and Autophagy

Apoptosis is a highly ordered and autonomous cell death process, which is regulated by B-cell lymphoma 2 (Bcl-2), the tumor suppressor gene P53, and a series of enzymes, including cysteine aspartic acid-specific protease. The caspase family and endonuclease G are regulated by the endogenous mitochondrial pathway, the exogenous apoptotic receptor pathway, and the endoplasmic reticulum pathway [7]. The use of large doses of GCs induces apoptosis of osteoblasts, prolongs the lifespan of osteoclasts, and reduces bone mineral density. The efficacy and side effects of GCs vary depending on the dose. In the human skeletal system, osteoblasts and osteoclasts have opposite effects, and there is a bidirectional signaling pathway between them. Their mutual balance is the basis for maintaining bone health. Schepper’s study showed that GCs can directly induce the apoptosis of osteoblasts, increase the activity of osteoclasts, inhibit the activity of osteoblasts, and consequently lead to the development of SANFH [42].

Recent research provides new insights into the inherent relationship between endothelial cell apoptosis, vascular injury, and SANFH. Specifically, GCs cause endoplasmic reticulum (ER) stress and induce apoptosis in endothelial cells, resulting in microvascular injury and ultimately leading to SANFH [43]. Nuclear receptor subfamily Group 3C member 1 (Nr3c1) is the primary receptor of GCs. Its downstream signaling pathway plays a crucial role in regulating physiological processes related to bone cells. Jiang’s study successfully created a zebrafish Nr3c1 mutant using CRISPR/Cas9 technology. This study confirmed that Nr3c1 mutation affected cartilage development and notably decreased the bone mineralized area [44]. In addition, the expression of genes associated with the extracellular matrix, osteoblasts, and osteoclasts was altered in Nr3c1 mutants. GCs regulate the expression of extracellular matrix-, osteoblast-, and osteoclast-related genes through NR3C1-dependent and NR3C1-independent pathways. This leads to osteoblast apoptosis, subsequently decreasing the load-bearing capacity of the femoral head and increasing the probability of SANFH.

Studies have shown that Dex significantly inhibits the proliferation and induces apoptosis of osteoblasts and mouse embryonic osteoblast precursor cells (MC3T3-E1) [45]. Yao points to the inhibition of induced kinase (PTEN) with an induced kinase inhibitor (VO-OHpic) to attenuate apoptosis and promote EPC angiogenesis in vitro. This is achieved by activating the transcription factor (Nrf2) signaling pathway and inhibiting mitochondrial apoptotic pathways. VO-OHpic can also increase angiogenesis in the femoral head. Both studies provide new possibilities for the treatment of SANFH [46]. Zhan demonstrated that allicin inhibits osteoblast apoptosis and the progression of SANFH by activating the PI3K/AKT pathway [47]. The extracellular vesicles of *Lactobacillus animalis* in the gut microbiota (GM) contain a variety of functional proteins. These proteins have the ability to promote angiogenesis, enhance bone health, and exhibit anti-apoptotic effects. However, these effects are notably diminished after the administration of GCs [48]. Apoptosis plays a key role in the progression of SANFH, and anti-apoptotic strategies may become important treatments for it.

Autophagy refers to the process by which cells degrade their own cytoplasmic proteins and damaged organelles. It is a mechanism of cell self-protection that safeguards the growth and development of cells, and regulates abnormal metabolism. However, long-term and high-dose glucocorticoid use alters the cellular environment, leading to cell stress and weakened autophagy. Failure of autophagy in clearing damaged organelles and harmful substances accelerates the imbalance of the femoral head microenvironment, leading to the occurrence of femoral head necrosis. As the metabolic center of cells, changes in the level of autophagy in the mitochondria will seriously affect the function of cells.

During the pathological process of SANFH, there will be an imbalance in bone remodeling, bone marrow mesenchymal stem cells’ self-osteogenic ability, and lipogenic differentiation ability. Osteoblasts and osteoclasts exhibit varying degrees of programmed cell death, and the functional imbalance and death of bone cells are all related to the level of mitochondrial autophagy [49,50]. Mitochondrial autophagy mainly plays its role in the following three ways: firstly, a ubiquitin adaptor protein binds to microtubule-associated protein 1 light chain 3 to form autophagosomes; secondly, putative kinase 1/ubiquitin ligase is induced by PTEN to bind autophagosomes to autophagosomes; finally, the damaged mitochondria are cleared through B-cell lymphoma 2 (Bcl-2)/adenovirus protein-interacting protein 3 (BNIP3)-nix and FUN14 domain 1 (FUNDC1) pathways to maintain intracellular homeostasis [51].

In an SANFH model, a high dose of GCs resulted in mitochondrial dysfunction and decreased autophagy levels. The ROS clearance ability is weakened, resulting in an increase in ROS levels, which induces the occurrence and regulation of BMSCs. The significantly increased contents of aging proteins such as P21 and P16 in BMSCs lead to stress aging in BMSCs, resulting in the occurrence and development of SANFH. Mitochondrial autophagy can be enhanced by regulating the related protein molecules P53 and Parkin in mitochondria, enabling BMSCs to effectively resist stress-induced apoptosis and senescence [52]. GCs inhibit the differentiation and mineralization of osteoblasts by down-regulating the expressions of autophagy and mitochondrial autophagy markers LC3-II, PINK1, and Parkin [53]. The increase in the overactivity of osteoclasts in SANFH is also an important factor contributing to its occurrence and development. The PARK2 protein is an essential regulatory protein in the process of mitochondrial autophagy. When autophagy levels decrease, overexpression of the PARK2 protein can promote the formation of osteoclasts (OC), while inhibition of the *PARK2* gene can significantly reduce the formation of OC [54]. At the same time, the loss of ULK1, a key factor in mitochondrial autophagy, weakens the ability of mitochondrial phagocytes, leading to the activation of the NLRP3 inflammasome, causing abnormal secretion of soluble cytokines, and subsequently promoting the differentiation and maturation of osteoclasts [55].

Therefore, the decrease in autophagy levels caused by GCs is one of the significant factors contributing to SANFH. The effect of GCs on autophagy leading to SANFH is illustrated in Figure 2.

### 3.5. Mechanism of Non-Coding RNAs

Non-coding RNA refers to RNA molecules that regulate gene expression and information transmission between cells but cannot encode proteins. They are also markers for potential disease diagnosis, collectively referred to as ncRNAs [56]. NcRNAs include microRNAs, lncRNAs, circRNAs, etc. In recent years, NcRNAs have become the focus of international medical research, and a large body of evidence indicates that the gene expression of NcRNAs is implicated in the onset and progression of SANFH. miRNA is a small fragment, single-chain, endogenous ncRNA that is involved in the occurrence and development of SANFH through the regulation of bone metabolism, BMSC proliferation and differentiation, and vascular repair. Wnt/β-catenin and TGF-β signaling pathways are classical signaling pathways of bone metabolism.

A substantial body of evidence indicates that the knockout of miR-145 can enhance the β-catenin pathway to suppress osteocyte apoptosis. Conversely, miR-141 can boost osteoblast activity and hinder osteoclasts through the TGF-β signaling pathway [57,58]. SPRY2 is the target gene of miR-122-5p, and inhibiting SPRY2 can stimulate osteoblast differentiation. MiR-122 may affect SANFH by regulating the SPRY2 gene [59]. miRNA-25-5p and miR-135b enhance the activity of nicotinamide adenine dinucleotide phosphate by activating the AMPK signaling pathway to counteract GC-induced oxidative stress in osteoblasts [60]. Upregulating miR-146a, WNT/FOXO, and Sirt1/NF-KB pathways stabilized osteoblast homeostasis to prevent SANFH [61]. Exosomes are nanoscale vesicles produced by cells, rich in various nucleic acids, proteins, and small molecules, with a diameter of around 30–150 nm [62]. Exosomes play an important role in immune response, cell proliferation, and nerve signaling. Exosomes play a crucial role in maintaining cell homeostasis, clearing cell debris, and promoting intercellular and interorgan communication by transporting proteins, lipids, RNA, and DNA. In addition, exosomes proliferate in all body fluids and transmit their molecular information in an autocrine, paracrine, and endocrine manner [63]. Significant progress has been made in utilizing exosomes to transport microRNA for the repair of SANFH. MiR-378-ASCs-Exosomes attenuate the development of SANFH by targeting fusion and activated sonic signaling pathways to enhance osteogenesis and angiogenesis [64]. MiR-26a-CD34-Exosomes protect the femoral head from GC-induced injury by enhancing angiogenesis and osteogenesis. The biological effects of MiR-26a-CD34-Exosomes can be utilized to prevent SANFH [65].

BMSCs are adult stem cells with robust proliferative capacity and the potential to differentiate in multiple directions. Under specific stimulation, BMSCs can be induced to differentiate into osteoblasts, chondroblasts, and other bone stromal cells. miR-155-5p can promote the cytoplasmic and nuclear translocation of β-catenin and activate the β-catenin expression signal, thereby enhancing the proliferation and osteogenic differentiation of BMSCs [66]. Overexpressed miR-15a-5p can significantly inhibit the expression of Wnt and β-catenin proteins and induce the production of the lipogenic transcription factor PPAR-γ, thereby inhibiting apoptosis of BMSCs, and promoting lipogenesis and osteogenic differentiation [67]. MiR-224-5p is up-regulated in GC-treated BMSCs. These results suggest that miR-224-5p can inhibit the osteogenesis of BMSCs and promote the adipogenic differentiation of BMSCs [68].

The silencing of miR-137-3p promotes osteogenesis and angiogenesis in vitro and in vivo. miR-137-3p silencing can upregulate Runx2 and CXCL12 to promote osteogenic differentiation, significantly increasing the number of EPCs. These genes may play a key role in SANFH repair [69]. Adipocyte-derived microvesicles containing miR-148a promote adipogenic differentiation by targeting the Wnt5a/Ror2 pathway and inhibit osteogenic differentiation [70]. MiRNAs with differential expression can regulate BMSCs through the same signaling pathway to some extent, thereby participating in the occurrence and development of SANFH. MicroRNA is differentially expressed in BMSCs, and this differential expression of microRNA plays a key regulatory role in the progression of SANFH. Studies have confirmed that microRNA CDR1as reduces the osteogenic effect of BMSCs and enhances their adipogenic differentiation ability through the miR-7-5p/WNT5B pathway. This leads to the occurrence of SANFH [71].

Up-regulation of microRNA-410 or down-regulation of Wnt-11 increases osteoblasts and decreases osteoclasts to relieve SANFH [72]. Long non-coding RNA (lncRNA) is an RNA sequence that is over 200 nucleotides long and possesses RNA-like characteristics. It was confirmed that LNCR-TMEM235 and BIRC5 mRNA can competitively bind miR-34a-3p to inhibit the apoptosis of BMSCs [73]. Additionally, ectopic overexpression of LNCR-EPIC1 can inhibit the hormone-induced apoptosis and programmed necrosis of OB-6 osteoblasts [74]. Lnc-ZFAS1 can affect the proliferation and osteogenic differentiation of osteoblasts. Although few studies have been conducted on lncRNA in SANFH, lncRNA still plays an important role in the pathogenesis of this disease as a potential therapeutic target for SANFH [75]. Circular RNA (circRNA) is a long endogenous circular RNA molecule with no protein-coding function. It participates in various genome transcriptions, maintains cell homeostasis, and regulates various physiological processes. Circ-0058792 can regulate the balance between osteogenesis and osteoclasts through the interaction of the TGF-β/Smad7 pathway with miR-181a-5p [76].

Additionally, circHIPK3 can promote the proliferation, migration, and angiogenesis of BMECs by targeting the miR-7 and KLF4/VEGF signaling pathways [77]. However, there is relatively little research on circRNA at present. From the above studies, it can be seen that ncRNA plays an important role in the pathogenesis of SANFH. Starting from ncRNA, we can gain a deeper understanding of the pathogenesis of SANFH and explore new targets for its treatment, aiming to prevent and treat SANFH at the genetic level.

### 3.6. Mechanisms of Genetic Susceptibility and Epigenetics

Genetic susceptibility refers to an individual’s increased or decreased sensitivity or vulnerability to a particular disease or disease-related factor at the genetic level. This susceptibility can be influenced by genetic variations in the genome, including Single-Nucleotide Polymorphisms (SNPs), gene copy number variations, gene mutations, etc. Due to differences in individual lifestyle, diet, surrounding environment, and genetic makeup, the microenvironment in each person varies. Therefore, the emergence of SANFH is evidently related to the individual’s hormonal tolerance dose. This difference may play a key role in the pathogenesis of SANFH. Zhao identified a single-nucleotide variant (*SNP rs311306*) in the *CR2* gene of Chinese Han males that was significantly associated with the risk of SANFH. The study confirmed that the *C* allele of SNP *rs311306* significantly increased the risk of SANFH through allele analysis. It provides clues to the nature of the etiological mechanism of SANFH [78]. The study found that the minor TG allele of *rs470154* in MMP10 was associated with an increased risk of SANFH. Additionally, MMP2 gene *rs2241146* and MMP10 gene *rs470154* were statistically correlated with an increased risk of SANFH, significantly increasing the probability of patients developing SANFH [79]. Therefore, genetic susceptibility plays an important role in the development of SANFH.

Epigenetic studies explore heritable changes in gene expression without altering the nucleotide sequence. They play an important role in growth, development, and disease evolution. At present, the primary research in SANFH includes DNA methylation, histone modification, RNA methylation, and other related areas. DNA methylation is a chemical modification that affects cell function by controlling gene expression and genome stability. It specifically refers to the process in which adenine or cytosine forms 5-methylcytosine (5mC) by covalently binding to methyl groups provided by S-adenosylmethionine (SAM) under the catalysis of DNA methyltransferase [80]. Current research on DNA methylation primarily focuses on CpG islands or short DNA fragments that are crucial for transcriptional regulation and abundant in high-CpG sites [81]. In recent years, genes such as *ABCB1, RUNX2, FZD1*, and *CARS*, as methylation sites, have increasingly matured in influencing the pathogenesis of SANFH through DNA methylation [82].

Osteoprotegerin (OPG) is primarily secreted by osteoblasts in bone tissue and belongs to the TNFR superfamily. The nuclear factor κB receptor activator (RANK) is synthesized by osteoblasts and bone marrow stromal cells. It is located on the surface of osteoclasts and osteoclast progenitors and belongs to the tumor necrosis factor receptor (TNFR) family. The transduction receptor RANKL is an essential cytokine for osteoclast differentiation. OPG can block the binding of RANKL and RANK through the competitive binding of RANKL, thereby inhibiting the activity and maturation of osteoclasts [83]. The OPG/RANK/RANKL system is the regulatory axis of osteoclasts and plays a crucial role in controlling bone metabolism [5]. Studies have shown that DNA methylation inhibits the expression of *OPG* and *RANKL* genes in the human skeletal system [84]. Hypermethylation of specific CpG sites increases the risk of SANFH. The methylation status of *OPG*, *RANK*, and *RANKL* genes in the serum of SANFH patients differs from that of the normal patient group. Detection of CpG site methylation of the aforementioned genes is beneficial for the early diagnosis of SANFH [82]. The TET family plays a key regulatory role in the process of DNA demethylation. In SANFH, 5hmC is upregulated, and the methylation level is influenced by TET to promote apoptosis of bone cells [85].

Histones, which bind to DNA to form nucleosomes, are structural proteins that constitute chromatin. Histone modifications mainly include acetylation, methylation, phosphorylation, and adenylation [86]. Histone deacetylases (HDACs) are enzymes that catalyze the deacetylation of histone proteins, facilitating the interaction between nucleosomes and the surrounding DNA, and they play a role in transcriptional silencing. They are mainly divided into four categories, among which the Sirtuins family belongs to class III HDACs, including SIRT1-7 [87]. Abnormal lipid metabolism plays a crucial role in the pathogenesis of SANFH, and the peroxisome proliferator-activated receptor γ (PPAR-γ) signaling pathway is essential for regulating the lipogenic differentiation of BMSCs [88]. Acetylation of histone H3K27 in the PPAR-γ promoter region is crucial for the development of SANFH. C/EBPα is a transcription factor that promotes lipogenic differentiation. In rat models of SANFH, C/EBPα contributes to the development of SANFH by inhibiting HDAC1, leading to an up-regulation of the acetylation level of histone H3K27 in the PPAR-γ promoter region. Inhibiting the histone acetylation level of PPARγ can effectively prevent lipogenic differentiation, thereby slowing down the progression of SANFH [89].

The expression of SIRT3 is decreased in BMSCs treated with GCs [90]. Promoting the expression of SIRT1 and SIRT3 will reduce cell apoptosis [91], while SIRT6 can prevent the occurrence of ONFH [92]. Serum HDAC9c-S levels in patients with SANFH are significantly up-regulated. The decrease in HDAC9 leads to reduced osteogenic and vasogenic capacity of BMSCs, and increased lipogenic capacity [93]. This suggests that the development of SANFH may be attributed to this enzyme’s role in regulating histone acetylation levels. Other studies have also indicated that CpG loci with a high methylation rate and high methylation level were found in CpG islands of *OPG, RANK,* and *RANKL* genes of SANFH patients. Hypermethylated CpG sites increase the risk of SANFH and provide a new target for the early prediction of SANFH occurrence [82].

### 3.7. Mechanism of Immune Imbalance

At present, many scholars believe that the dynamic balance between bone formation and bone resorption is a regulated inflammatory response, and various inflammatory cells and cytokines play a regulatory role in maintaining bone homeostasis [94]. A significant feature of SANFH is the presence of an abnormal chronic inflammatory response [94]. Long-term exposure to harmful stimuli and subsequent disruption of the bone tissue microenvironment are important factors in triggering chronic inflammation [95]. Under chronic inflammatory conditions, the process of effective reconstruction and repair of necrotic bone is impeded [96]. The immune system is the body’s primary defense against infection and disease.

It maintains a state of equilibrium through a complex series of signaling pathways and cellular responses. Immune imbalances, such as the overactivation or suppression of the inflammatory response, can lead to a variety of diseases. Studies have shown that hormone therapy can trigger an abnormal immune system response, leading to injury to the blood vessels of the femoral head and necrosis of bone tissue [8,97,98]. In particular, long-term or high-dose hormone use can lead to local and systemic inflammatory responses by activating multiple inflammatory cells and releasing inflammatory mediators. TLR4 is a crucial receptor in the immune system that detects pathogens and signals of damage. It was found that the expression of TLR4 and its downstream signaling molecules, such as MyD88 and NF-κB, was increased in an SANFH model, suggesting a potential role of the TLR4 signaling pathway in the pathogenesis of SANFH [97]. Type 1 macrophages promote inflammatory responses, while M2 macrophages are involved in inflammation resolution and tissue repair.

In SANFH, the M1/M2 ratio is unbalanced, leading to persistent inflammation and increased tissue damage. Targeting the mechanism of immune imbalance in SANFH, the development of targeted immunoregulatory treatment strategies may be an effective approach to treatment. For example, by modulating specific inflammatory signaling pathways or adjusting the polarization state of macrophages, it may help reduce the inflammatory response, promote tissue repair, and halt or delay the progression of SANFH.

## 4. Discussion

SANFH is a refractory and disabling joint disease. The main population of SANFH patients consists of patients who have been using high doses of glucocorticoids for an extended period. SANFH can significantly impair patients’ ability to work and impose a substantial economic burden on both the family and society. Therefore, for the early diagnosis and treatment of SANFH, it is crucial to use medication judiciously to delay the progression of SANFH. However, there is a lack of specific drugs for the treatment of SANFH; therefore, it is urgent to explore the pathogenesis of SANFH.

Studies have shown that drugs promoting angiogenesis, such as ICA, Huoxue Tongluo, and VO-OHpic, are potential therapeutic agents for SANFH [46,99]. At present, Chinese herbal extracts, such as astragaloside, resveratrol, hesperidin, and icariin, have positive effects on osteogenesis and the treatment of SANFH [99,100,101,102]. Exosomes are nanoscale vesicles produced by cells with a diameter of around 30–150 nm. They are rich in various nucleic acids, proteins, and small molecules from their source cells. Exosomes transport proteins, lipids, mRNA, miRNA, lncRNA, and DNA. They play an important role in maintaining cell homeostasis, clearing cell debris, and promoting intercellular and interorgan communication [103]. Exosomes derived from BMSCs have a promising potential in promoting osteogenic differentiation and improving SANFH [104]. In the realm of drug delivery materials, including tetrahedral DNA, hydrogels, and nanoparticles, targeted delivery of bone drugs can enhance bone formation, providing a solid basis for the clinical treatment of SANFH. However, there are still some limitations in these studies [105,106].

## 5. Conclusions

In this review, we describe and summarize the mechanisms that contribute to the occurrence and progression of SANFH. SANFH is a complex process involving multiple mechanisms, factors, and signaling pathways. Currently, the academic community believes that the theory of “multiple strikes” is more consistent with the pathogenesis of SANFH. A single mechanism is insufficient to explain the complexity of its pathogenesis and the challenges associated with its treatment. While clarifying the mechanism of occurrence and development of SANFH, effective measures should be taken to intervene in order to maximize the curative effect for SANFH and improve the quality of life of patients. With the advancement of modern diagnostic technology and the rapid progress of molecular medicine, it is anticipated that in the near future, precise gene-level targeting therapy can be implemented to address the pathogenesis of SANFH. This will enable early precision treatment of SANFH, alleviate patient suffering, and provide significant benefits to patients.

## Figures and Tables

**Figure 1 biomolecules-14-00667-f001:**
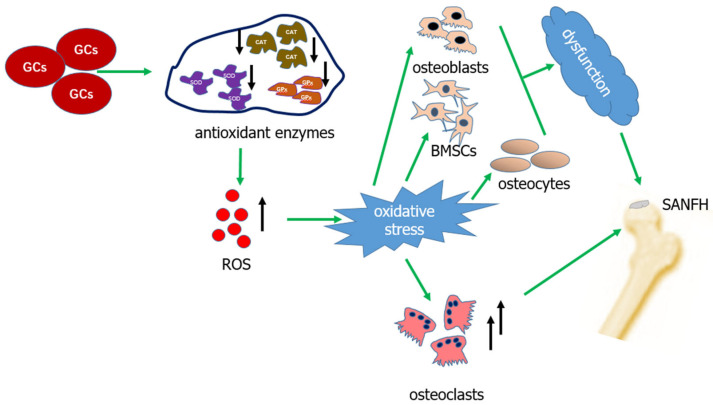
The mechanism by which GCs induce oxidative stress, leading to SANFH.

**Figure 2 biomolecules-14-00667-f002:**
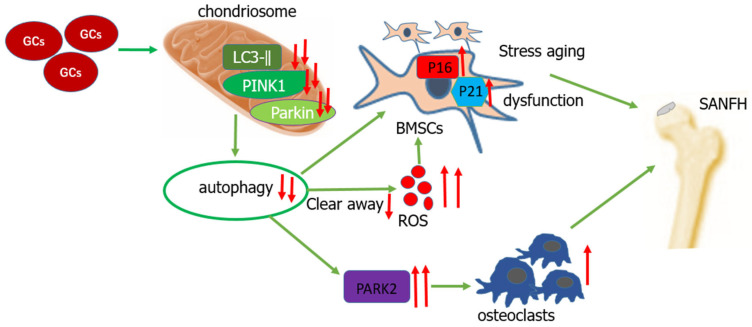
The mechanism by which glucocorticoid-induced reduction in autophagy leads to SANFH.

## Data Availability

Data are contained within the article.

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
