# Peer review of "Advances in the Pathogenesis of Steroid-Associated Osteonecrosis of the Femoral Head"

_biomolecules, 2024, doi:10.3390/biom14060667_

Round 1

Reviewer 1 Report

Comments and Suggestions for Authors

The manuscript contains large portion of the theoretical knowledge based on the latest  publications. I recommend that you consider changing the article to a literature review.

  Comments on the Quality of English Language

The 6th sentence:

An epidemiological survey shows that the annual prevalence of ONFH in Japan is 18.2-19.2 per 100,000 people [2]. The incidence of ONFH in China is as high as more than 8 million people aged 15 and above.

It would be easier to campare: Japan is 18.2-19.2 per 100,000 people and China ..... per 100,000 people.

The 8th sentence

SANFH instead of SNAFH

In conlusions:

The first sentence:

In this review, we summarize the mechanisms that contribute to the occurrence and progression of SANFH and describe how various mechanisms contribute to the occurrence of SANFH.

I would change to :

In this review, we describe and summarize the mechanisms that contribute to the occurrence and progression of SANFH.

Reviewer 2 Report

Comments and Suggestions for Authors

This review is very insightful and also comprehensive to understand how steroid-associated osteonecrosis of the femoral head by various physiological and cellular mechanism. The review is well-written in logic and step-wise approach to provide the knowledge. The strength of this review is that it gives the professional and special information to the scientists in this area as well as even the researchers in other area and normal people who are interested in this subject can understand and is useful for them. 

The English is well-written almost no need to edit, apart from two minor points: 

1. On Figure 1, GC in blue circle at the starting point is too small to read. 

2. At 2.3 Lipid metabolism..., on 2nd line, bone marrow stromal cells -> bone marrow stromal cells (BMSCs) (reason : BMSCs is on Figure 1) 

Reviewer 3 Report

Comments and Suggestions for Authors

 Although this manuscript is well written, some points are insufficient.

1) Page4; Bone mass is determined by the balance between osteoblastic bone formation and osteoclastic bone resorption. Explain this. Explain BMP-2 signaling and RANKL signaling.

2) Page7; miRNAs and microRNAs are the same. Delete miRNAs.

3) Page8; Explain Exos.

Round 2

Reviewer 3 Report

Comments and Suggestions for Authors

Page8; Explain exosomes in the main body, although you described exosomes in the discussion section.

Author Response

Thank you very much for taking the time to review this manuscript. Thank you very much for your suggestions. I have included the pertinent information about exosomes and made revisions to the article.

Comment1:Page8; Explain exosomes in the main body, although you described exosomes in the discussion section.

Response 1: Exosomes are nanoscale vesicles produced by cells, rich in various nucleic acids, proteins, and small molecules, with a diameter of about 30-150 nm [62]. Exosomes play an important role in immune response, cell proliferation, and nerve signaling. Exosomes play a crucial role in maintaining cell homeostasis, clearing cell debris, and promoting intercellular and interorgan communication by transporting proteins, lipids, RNA, and DNA. In addition, exosomes proliferate in all body fluids and transmit their molecular information in an autocrine, paracrine, and endocrine manner [63]. Significant progress has been made in utilizing exosomes to transport microRNA for the repair of SANFH.